# Evolution of cardiac tissue and flow mechanics in developing Japanese Medaka

**Sreyashi Chakraborty[1], Sayantan Bhattacharya[2], Brett Albert Meyers[1], Maria S. Sepúlveda[3], Pavlos P. Vlachos[1,4]***

**1** Department of Mechanical Engineering, Purdue University, West Lafayette, Indiana, United States of America, **2** Department of Mechanical Engineering, University of Maryland, Baltimore County, Baltimore, Maryland, United States of America, **3** Department of Forestry & Natural Resources, Purdue University, West Lafayette, Indiana, United States of America, **4** Department of Biomedical Engineering, Purdue University, West Lafayette, Indiana, United States of America

\* pvlachos@purdue.edu

## Abstract

The effects of pressure drop across cardiac valve cushion regions and endocardial wall strain in the early developmental stages of a teleost species heart are poorly understood. In the presented work, we utilize microscale particle image velocimetry (μPIV) flow measurements of developing medaka hearts from 3 to 14 dpf (n = 5 at each dpf) to quantify the pressure field and endocardial wall strain. Peak pressure drop at the atrioventricular canal ($\Delta P_{AVC}$) and outflow tract ($\Delta P_{OFT}$) show a steady increase with fish age progression. Pressure drops when non-dimensionalized with blood viscosity and heart rate at each dpf are comparable with measurements in zebrafish hearts. Retrograde flows captured at these regions display a negative pressure drop. A novel metric, Endocardial Work (EW), is introduced by analyzing the $\Delta P_{AVC}$-strain curves, which is a non-invasive measure of work required for ventricle filling. EW is a metric that can differentiate between the linear heart stage (< 100 Pa-%), cardiac looped chamber stage (< 300 Pa-%), and the fully formed chamber stage (> 300 Pa-%).

**Data Availability Statement:** All raw and postprocessed data files are available from the https://doi.org/10.6084/m9.figshare.26390866.

## Introduction

Cardiac chamber formation and development in teleosts is a dynamic process where mechanical forces generated by the pulsating heart cause endocardial cells to migrate or reorganize, leading to cardiac remodeling [1]. However, past studies [2,3] investigating biomechanics have mainly probed into the flow wall shear stress (WSS) variation, with little to no attention to the pressure drop (ΔP) that induces the varying WSS profiles or the cardiac wall strain facilitating morphological changes in the endocardium. Pressure and strain data can potentially indicate the amount of endocardial work (EW) required for ventricle filling or ejection [4–9]. Standard indices of ventricle function do not measure myocardial work related to metabolic demand. The EW has been shown in humans and higher vertebrates to directly reflect oxygen demand and metabolic consumption [4]. Hence it is a comprehensive metric for regional ventricle function that can be developed to distinguish between healthy and diseased ventricles [4,5]. To

**Funding:** The author(s) received no specific funding for this work.

**Competing interests:** The authors have declared that no competing interests exist.

our knowledge, no studies have evaluated in-vivo EW in teleost ventricles. Owing to the small size of the teleosts and limited spatial resolution of images, it is challenging to accurately calculate velocity gradients that contribute to calculations of pressure and strain.

Pressure measurements in previous studies were time-consuming. They were done with cannulae at a single point, with intrusive probes that provided limited information about the spatiotemporal pressure variation [10,11]. Pressure gradients in the zebrafish (*Danio rerio*) heart were calculated using Laplace law [12] or from the knowledge of circumferential stress [13]. However, the accuracy of these measurements is limited since they neglected viscous forces in the heart. A non-invasive spatially and temporally resolved pressure field in a 48-hour post-fertilization (hpf) zebrafish heart was first calculated using micro-Particle Image Velocimetry (μPIV) analysis by Dasi et al. [14]. The imaging was limited to the atrium only because the opacity of fish's eyes and head blocked the visibility of a part of the ventricle.

Cardiac wall strain measurements of teleosts are affected by spatially under-resolved images that cannot accurately capture the wall motion. The recent introduction of high-frequency ultrasound probes designed for small animals enabled Hein et al. [15] to measure strain rate and strain (%) measurements of adult zebrafish suffering a myocardial injury. However, echocardiographic measurements are impossible to acquire from embryonic and larval fish due to the small size and difficulty of introducing a contrast agent. Lee et al. utilized the optical transparency of 48 hpf, 75 hpf, and 100 hpf zebrafish (both control and mutants) and used selective plane illumination microscopic imaging technique to report strain measurements [3].

Thus, an overview of the literature highlights the need to measure pressure and strain in teleost hearts accurately. To our knowledge, there is no baseline framework for embryonic vertebrate animal models that investigates the variation of EW along the developmental stages. In the current work, we selected Japanese Medaka (*Oryzias latipes*) as our teleost model because it is larger than zebrafish in both embryonic and larval stages. Additionally, medaka grows two times slower than zebrafish [16,17] allowing capturing developmental landmarks during cardiac morphogenesis from embryonic to larval stages.

The objective of this work is to conduct a series of live experiments to image the Japanese medaka heart under a microscope followed by a subsequent μPIV analysis and quantify the ΔP across the atrioventricular canal (AVC) and outflow tract (OFT), endocardial wall strain of the ventricle and consequently the EW required for ventricle filling across development in embryo/larva.

## Materials and methods

Details about the Japanese Medaka Husbandry, Data Acquisition, and μPIV analysis in the medaka heart is discussed in the previous work using this data [18] but also added here for comprehensiveness. The fish maintenance and experimental protocols were approved by Purdue Animal Care and Use Committee (PACUC) (Protocol Number 1702001545). The study was performed in accordance with the ARRIVE guidelines. All methods were performed in accordance with relevant guidelines and regulations as approved by PACUC.

### Japanese Medaka Husbandry and embryo collection

Wild type adult Japanese medaka of an inbred SK2MC strain (20 male-female pairs) obtained from USEPA, Duluth, MN, USA were mated in 2L transparent tanks with continuous oxygen supplied by an aquarium pump under artificial reproductive conditions (14:10; Light:Dark cycle; 26–28°C). Dissolved oxygen level was always above 5 mg/L, pH was between 7–8, and total ammonia levels were below 0.25 ppm. The fish spawned eggs every day at 9 am in the morning and eggs were fertilized by 5 pm in the evening. The fertilized eggs were collected,

immersed in embryo medium (50% diluted saline solution containing 2 ppm methylene blue) and stored in an environment chamber at 28°C. Imaging was performed after the onset of blood circulation in the embryos at 3 dpf. Chorions were intact with the embryos and image distortions were assumed to be negligible due to the presence of chorions.

## Fish selection for imaging

A total of 150 fertilized eggs were collected and five embryos/larvae (n = 5) were imaged each day starting at 3 dpf until 12 dpf. The temperature was kept constant by placing the samples in a chamber whose temperature was between 26°C and 28°C. On average, hatching occurred between 8–9 dpf. Larvae were anesthetized using MS-222 (226mg/L) solution to prevent movement during imaging. Embryos were collected from the population by random sampling.

## Data acquisition

Images of medaka fish heart were collected from animals ranging from 3 to 14 dpf (n = 5 at all-time points). They were visualized through a Nikon-Ti microscope with a 60x objective lens and 40x objective lens. The sample was positioned on a 3-axis translation stage (Aerotech: x-resolution = 1 μm, y-resolution = 1 μm, z-resolution = 0.1 μm). The images were acquired by a high-speed CMOS camera (Phantom Miro-310, 2500x1600 pixels) which resulted in an effective pixel size of 0.16 μm (for 60x) and 0.25 μm (for 40x). Brightfield time-resolved images were captured for 4s at a rate of 400 frames per second. These correspond to 8–10 cycles and around 199 image pairs per cycle.

## Flow velocity analysis in heart

**Signal amplification of raw images.** The raw images (Fig 1A) were preprocessed by applying a proper orthogonal decomposition [19–22] across time series of images and retaining the non-dominant modes that represent the fluctuations of red blood corpuscle (RBC) patterns across time (Fig 1B). This step substantially removed the background artifacts. The RBC movement patterns across time lapse images were cross-correlated and a phase average of the cross-correlations were used to obtain velocities in the heart and vessels as shown in Fig 1C.

**Heart function measurement.** The heart rate (HR) was measured from the Fourier transform of the instantaneous velocity waveforms. The Reynolds number ($Re = \frac{\rho V d}{\mu}$) and Womersley number ($Wo = d\sqrt{\frac{\rho\omega}{\mu}}$) were calculated for each sample and each age based on the atrial

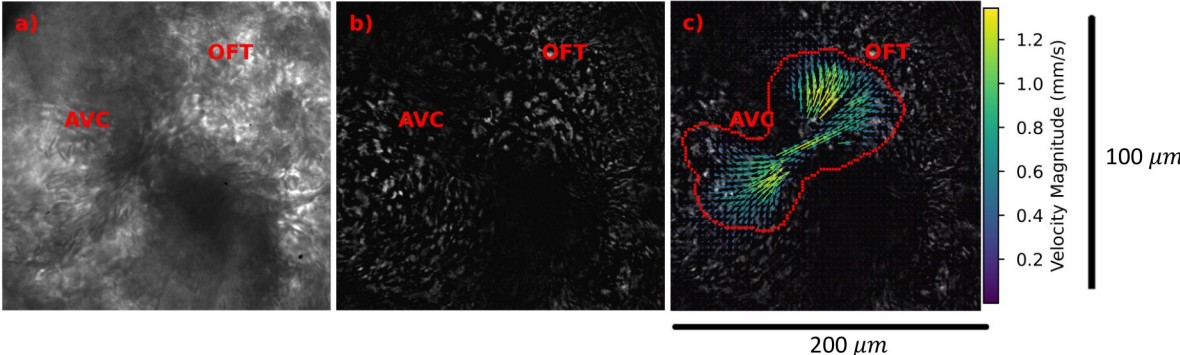

**Fig 1. Raw images to velocity vector fields.** a) raw image b) signal amplified image and c) velocity vectors superimposed on raw image of a 13 dpf medaka heart during ventricle diastole. AVC = Atrioventricular canal; OFT = ventricle outflow tract. The color of the vectors corresponds to the velocity magnitude as shown by the colorbar. The scalebar in **μ**m is shown on the right side.

inlet diameter (d), blood density ($\rho$) as 1025 kg/m$^3$ [23], blood dynamic viscosities ($\mu$) that change with dpf and $\omega = 2\pi HR/60$. The ejection fraction (EF) is a function of ventricle volumes at diastole and systole. A is the ventricle area at peak ventricle diastole. It is defined in Eq (1). L is the length of the ventricle and $D_1$, $D_2$ are breadth and depth of the ventricle respectively.

$$A = \pi L D_1; \; Vol = \frac{\pi}{4} L D_1 D_2; \; D_1 = D_2;$$

$$EF = \frac{end\ diastolic\ volume - end\ systolic\ volume}{end\ diastolic\ volume} \tag{1}$$

Following Bulk et al. [13], a cylindrical area and volume are assumed for the ventricle due to limitations in resolving depth in a planar micro-PIV measurement. However, the assumption is restrictive as the ventricle deviates from tubular shape during development.

## μPIV analysis in heart

The instantaneous velocity field at a time point in one pulsatile cycle was obtained from the standard cross correlation (SCC) of two successive image pairs as mentioned previously [24]. Each image region is subdivided into windows before cross-correlating. Cross-correlation vectors were obtained using an initial window size of 96x96 pixels (window resolution: 48x48 pixels) with a 50% overlap between windows, followed by another pass of window size of 64x64 pixels (window resolution: 32x32 pixels) with 50% overlap and finally followed by another pass of window size of 64x64 pixels (window resolution: 32x32 pixels) with 75% overlap [25]. Each window in each image in the first pass, second and third pass has roughly 10–15 RBC patterns, 6–8 RBC patterns, and 5–7 RBC patterns respectively. In each pass, an iterative window deformation technique (with Blackman filter) was applied with a minimum 3 and maximum 6 iterations [26,27]. This minimized any loss of correlation due to velocity gradients, out-of-plane motion intensity dropout and RBC patterns crossing the correlation window. The correlation planes at each phase of the cycle were averaged across all cycles to get an ensemble correlation plane [28], as demonstrated in S1 Fig. The peak location of the correlation plane denotes the displacement between images at that time point. An accurate velocity estimate at each cycle phase is obtained by fitting sub-pixel resolution curves to the correlation peak using a three point Gaussian estimator [24,29].

The spatial resolution of the final phase averaged velocity map is 1.28 μm for 60x images and 2 μm for 40x images. The temporal resolution is 2ms. No temporal or spatial averaging was done to the velocity measurements since the peak flow in the atrioventricular canal (AVC) and the outflow tract (OFT) were changing very rapidly across time and space. Universal outlier detection was done in between passes to get rid of erroneous velocity vectors [30]. If a normalized fluctuation based on the neighborhood (3x3 vector grid) median was greater than a threshold of 2, the vector was identified as an outlier. An additional median filtering with a window size of 3x3 pixels spatially was done to the vectors in the final pass [31,32]. All the calculation steps are done using the in-house PIV software, PRANA [33].

## Velocity measurement location in medaka

Fig 2A shows the side view of a larval medaka of age 14 dpf where the location of heart, dorsal vessels and caudal vessels are highlighted. The AVC and OFT regions are represented by a red rectangle and a green rectangle respectively in parts b), c) and d). In Fig 2B, a schematic of the linear heart tube during 3 dpf to 5 dpf is shown. The AVC region is wider than the OFT region.

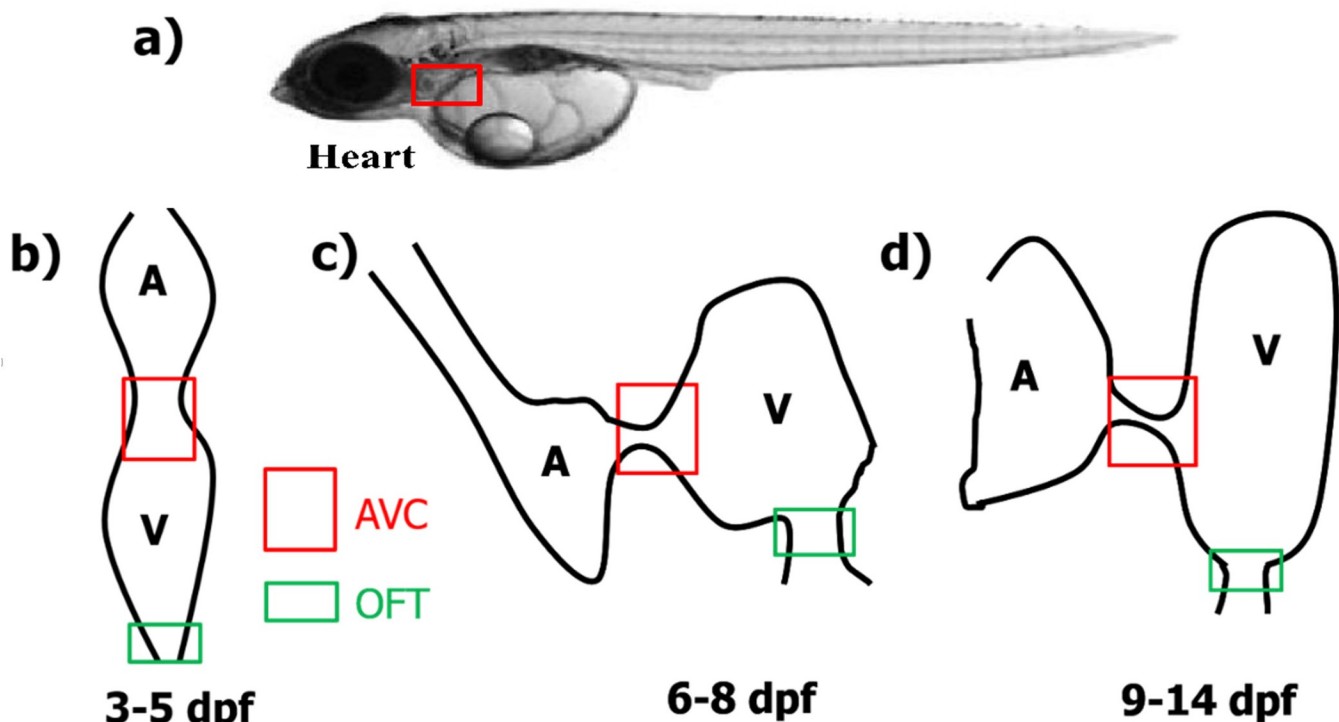

**Fig 2. Location of heart and vessels in medaka and schematic diagram of medaka heart at different ages.** (a) Location of heart when viewed from the side in a larval medaka (age 14 dpf). Location of AVC and OFT in a schematic medaka heart during (b) 3 dpf to 5 dpf, (c) 6 dpf to 8 dpf and (d) 9 dpf to 14 dpf. "A" represents atrium and "V" represents ventricle. Red rectangle represents location of measurement at AVC and green rectangle represents location of measurement at OFT in (b), (c) and (d). Fig 2B–2D are representative sketches based on the author's visual observations during experiments.

The atrio-ventricular flexion at the end of cardiac looping has moved atrium and ventricle in different planes in Fig 2C. The AVC is narrower than the OFT in this stage from 6 dpf to 8 dpf. In Fig 2D both AVC and OFT has further reduced in cross-section while the atrium and ventricle have grown in size.

In the linear heart tube stage, the atrium and ventricle are in line and on the same plane as shown in Fig 2B. A contraction wave starts at the atrial inflow and propagates along the entire length of the heart tube with uniform speed. The contraction wave squeezes different cross sections of the tube sequentially, pushing blood out of the heart into the dorsal aorta (DA) without significant backflow despite the lack of valves at this stage [34].

## Velocity reconstruction across the AVC and the OFT

The medaka heart has two small connector regions that reduce the area as the fish ages from 3 dpf to 14 dpf. One is the AVC [diameter: 6–10 μm] joining the atrium and ventricle, and the other is the OFT [diameter: 10–14 μm] connecting the ventricle and the BA. The high velocities at AVC and OFT at certain heart cycle phases could not be accurately captured due to increased noise or signal dropout in those frames. So, using the velocity information of a neighboring region in the ventricle/atrium and applying the conservation of mass principle as shown in Eq (2) the velocity vectors were reconstructed in these areas.

$$\nabla . \overrightarrow{u} = 0 \tag{2}$$

## Velocity gradient calculation

Velocity gradients were calculated using a high accuracy gradient calculation technique namely the Compact-Richardson method [35] in accordance to second order polynomial fitting done in previous studies [2,14]. Shear rate ($\gamma$) at the AVC and OFT were derived from the velocity gradient field using the following equation.

$$\gamma = \frac{d\overrightarrow{V}}{d\overrightarrow{n}} \tag{3}$$

where $\overrightarrow{V}$ is the flow velocity vector, $\overrightarrow{n}$ is the unit vector normal to the cardiac wall surface and $\frac{d\overrightarrow{V}}{d\overrightarrow{n}}$ is the directional derivative of the velocity in the direction normal to the cardiac wall surface.

Blood dynamic viscosity in teleosts is ill-defined. We modelled the medaka heart blood viscosity as a function of both hematocrit (Ht) and wall shear rate ($\gamma$) according to the Walburn-Schneck equation as shown in [36].

## Hematocrit (Ht) calculation

Ventricles images were cropped to retain a small area around the ventricle center. RBC pattern count (Nrbc) was calculated from the average auto-correlation plane of all the time-lapse ventricle images for each medaka heart as discussed in [37]. The ventricle volume ($V_{vol}$) was calculated by multiplying the area of the images with the depth of focus. Then the Ht, defined by the percentage by volume of RBC present in the blood, was calculated using Eq (4).

$$Ht = 100 * Volume\ of\ a\ RBC * \frac{N_{rbc}}{V_{vol}} \tag{4}$$

## Dynamic viscosity ($\mu$) calculation

Viscosity at each dpf for each sample is needed in the pressure calculation equation. The Walburn-Schneck model for a non-Newtonian fluid [36,38] is used to calculate blood viscosity ($\mu$) as a function of hematocrit (Ht) and shear rate ($\gamma$). In this work, the peak shear rate recorded in a heart cycle and the Ht in percentage calculated from Eq (4) is used in Eq (5).

$$\mu = C1 \exp(C2*Ht) * \exp\left( C4\left(\frac{TPMA}{Ht^2}\right)\right) * \gamma^{-C3*Ht} \tag{5}$$

Here, C1 = 0.00797, C2 = 0.0608, C3 = 0.00499, C4 = 14.59 L/g (38), and total protein minus albumin (TPMA) = 45 mg/mL [39].

## Pressure analysis of the cardiac chambers

The pressure field in the cardiac chambers for a heart cycle was obtained by integrating Eq (6) for the known phase averaged velocity field. Velocity gradients were calculated using a high accuracy gradient calculation technique, namely the Compact-Richardson method [35] in accordance with second-order polynomial fitting done in previous studies [2,14].

$$\nabla P = -\rho\left(\frac{\partial\overrightarrow{u}}{\partial t} + \overrightarrow{u}.\nabla\overrightarrow{u}\right) + \mu\left(\nabla^2\overrightarrow{u}\right) \tag{6}$$

The pressure was then integrated along multiple line paths in the cardiac field, assuming a zero value at a fixed node on the boundary. Integration paths were averaged to smooth out

uncertainties in the velocity field. The average value was updated in successive iterations until the pressure residual converges to a value of 1x $10^{-3}$. A second-order trapezoidal rule was used for spatial integration. The dependence of this omnidirectional method on grid resolution, sampling rate, velocity field error levels, and off-axis recording was explored extensively for PIV measured velocity fields in previous work from our lab [40]. Blood dynamic viscosity (μ) was calculated for each sample at each dpf using the hematocrit and shear rate information as described in previous work [18]. Owing to small Re (<<1) and Wo (<<1), the first two terms on the right-hand side of Eq (6) are negligible. So pressure (P) can be non-dimensionalized by the product of heart rate and blood dynamic viscosity (HR*μ) as shown below.

Eq (6) denotes a rearranged form of the unsteady Navier Stokes equation for incompressible flow. We denote a representative length scale, time scale and velocity scale of the flow by $L$, $T$, and $U$ respectively where $U = L/T$. To deduce the correct non-dimensionalization for pressure from Eq (6), we subsitute $L$, $T$, $U$, $P$ and do an order of magnitude analysis of the individual terms to obtain Eq (7).

$$\frac{P}{L} = \rho\left(\frac{U}{T}\right) + \rho\left(\frac{U^2}{L}\right) + \mu\left(\frac{U}{L^2}\right) \tag{7}$$

Multiplying by (LT/μ), we get,

$$\frac{P}{\left(\frac{1}{T}\right)*\mu} = \rho\left(\frac{UL}{\mu}\right) + \rho\left(\frac{U^2 T}{\mu}\right) + \mu\left(\frac{UT}{L\mu}\right) \tag{8}$$

As $U = L/T$, the first two terms on right side of Eq (8) can be simplified to Re. For creeping flow (Re<<1), thus, these terms become negligible. The third term on the right side simplifies to 1. Hence, the pressure term should be non-dimensionalized by $\left(\frac{1}{T}\right)*\mu$. Since, heart rate has dimensions $\left(\frac{1}{T}\right)$, we get the dimensionless equation for pressure as,

$$\frac{P}{HR*\mu} = 1 \tag{9}$$

Considering the dimensions of individual terms, $P = M/(LTT)$, $\mu = M/(LT)$, and $HR = 1/T$, we can see that Eq (9) is dimensionless.

## Strain analysis of the ventricle wall

Velocity field measurements are used to calculate the deformation gradient tensor F for the entire ventricle. The end of ventricle systole is used as the reference time (or frame number) against which the deformation of the ventricle at other time points in the heart cycle is calculated. Though ECG measurements [41] could inform a precise reference time, it was out of the scope of the experiment. Multi-cycle recording was helpful in establishing an average cycle time and each cycle was checked with respect to the reference point and cycle time (199 image pairs) for consistency. The Strain is then calculated from F according to the equation below

$$F = \begin{bmatrix} \frac{\partial u}{\partial x} & \frac{\partial u}{\partial y} \\ \frac{\partial v}{\partial x} & \frac{\partial v}{\partial y} \end{bmatrix}; \; Strain = \frac{1}{2}\left(F^T F - I\right) \tag{10}$$

Where T denotes the transpose of a matrix, and I is the identity matrix [42]. The endocardial wall location was identified using an automated segmentation on the velocity gradient magnitude. Previous micro-PIV application [43] used a threshold on velocity magnitude

isosurface to identify OFT wall location. The complex wall contour for this measurement was more consistently estimated by accounting for the velocity graident. The velocity magnitude obtained from the PIV processing is used to evaluate the gradients using discrete difference with Compact-Richardson method. To eliminate noisy gradients in the non-flow region, the gradient field magnitude is multiplied with velocity magnitude. The product field is filtered to retain high values above a threshold of 90th percentile. This filtered gradient field is binarized using Otsu's method in MATLAB. To minimize discontinuities in connected components, MATLAB erosion dilation (a disk element of radius 3 pixels) and imfill (to fill holes with 8-point edge connectivity) operations were applied on the filtered field. The connected component with maximum number of pixels was selected as the mask. The boundary of this mask denotes the wall location. Finally, the estimated Lagrangian strain field values on the identified boundary points were indicated by a colormap to show the spatial variation of strain along the ventricle wall. At each time point, three maximum strain values were further extracted and averaged to generate peak endocardial wall strain variation along a cardiac cycle. Variation of these three maximum values will not be significant, given that a sufficient number of points (much greater than 50 entries; density varies by case) exists along the border with a resolution of 1.3 μm to 2.0 μm between boundary points. This narrow spacing ensures large rate of change of transmural strain between neighboring boundary points will not exist.

## EW measurement of the ventricle

The endocardial work (EW) is measured by calculating the area under the curve of a $\Delta P_{AVC}$-strain plot. The area calculation is performed in Matlab by implementing a numerical trapezoidal integration technique. The $\Delta P_{AVC}$-strain plot is not a closed loop because the $\Delta P$ information is available only for the diastolic cycle of the ventricle when the AVC is open. The EW measure reflects the combined effect of changes in $\Delta P_{AVC}$ and strain on the effective contractility of the endocardial wall.

## Results

### Cardiac function and geometry

The progression of cardiac function and geometry parameters for this dataset have been described in detail in our previous analysis of wall shear stress (WSS) measurements done with this data (n = 5 for each parameter at each dpf) [18]. The mean value of each parameter is

**Table 1. Cardiac function and geometry parameters.**

| dpf | HR (bpm) | Re | Wo | A (μm²) | EF (%) |
|---|---|---|---|---|---|
| 3 | 78.5 ± 3.7 | 0.02 ± 0.003 | 0.021 ± 0.005 | 1143.3 ± 75.9 | 58.2 ± 4.7 |
| 4 | 85.5 ± 3.3 | 0.03 ± 0.007 | 0.022 ± 0.004 | 1433.3 ± 49.9 | 58.2 ± 7.9 |
| 5 | 93 ± 6.5 | 0.03 ± 0.008 | 0.023 ± 0.006 | 1610 ± 151.2 | 72.2 ± 4.8 |
| 6 | 93 ± 2.4 | 0.04 ± 0.005 | 0.023 ± 0.004 | 1466.7 ± 212.5 | 54.5 ± 7.4 |
| 7 | 106 ± 7.5 | 0.04 ± 0.008 | 0.024 ± 0.006 | 1857.5 ± 369.3 | 50.5 ± 9.3 |
| 8 | 132 ± 4.9 | 0.04 ± 0.003 | 0.027 ± 0.005 | 1604 ± 104.8 | 59.5 ± 10 |
| 9 | 150.5 ± 2.5 | 0.06 ± 0.011 | 0.029 ± 0.004 | 2660 ± 298.5 | 61.2 ± 11.9 |
| 10 | 143.3 ± 6.2 | 0.07 ± 0.009 | 0.028 ± 0.006 | 4065 ± 648.5 | 81.3 ± 3.4 |
| 11 | 161.3 ± 18.7 | 0.08 ± 0.027 | 0.029 ± 0.010 | 3796.7 ± 369.2 | 72.6 ± 3.1 |
| 12 | 145 ± 2.4 | 0.07 ± 0.007 | 0.028 ± 0.004 | 3531.2 ± 457.5 | 75.9 ± 6.7 |
| 13 | 173.6 ± 7.1 | 0.1 ± 0.034 | 0.031 ± 0.006 | 4884 ± 1095.5 | 76.3 ± 4.8 |
| 14 | 172.4 ± 7.1 | 0.1 ± 0.008 | 0.031 ± 0.006 | 3906 ± 1312.3 | 81 ± 5.5 |

presented in **Table 1**. HR represents heart rate in beats per minute (bpm), Re and Wo represent Reynolds number and Womersley number, respectively, and both are non-dimensional. "A" represents the ventricle area during peak diastole. EF represents the ejection fraction of the ventricle.

### Pressure contours across the AVC and the OFT

Pressure field variation across the atrium and ventricle along three phases of the ventricle filling cycle i.e., diastole is plotted in Fig 3 for a medaka heart at 3 dpf, 8 dpf, and 13 dpf. A zero-pressure boundary condition is applied at the ventricle outflow. Both the atrial inflow and the ventricle outflow remain closed throughout $T_{dias}$, which is the period of the ventricle filling cycle. At $0.3T_{dias}$ the peak pressure drop across the AVC is $800*(HR*\mu)$ at 3dpf, $1200*(HR*\mu)$ at 8 dpf,

and $2000*(HR*\mu)$ at 13dpf. The atrium experiences the highest pressure, and blood flows through the AVC from the atrium to the ventricle. At $0.6T_{dias}$ the AVC pressure drop reduces in magnitude while blood continues to flow in the same direction [$500*(HR*\mu)$ at 3dpf, $900*(HR*\mu)$ at 8 dpf and $1200*(HR*\mu)$ at 13dpf]. At $0.9T_{dias}$ the ventricle filling has almost ended with blood flowing from atrium to ventricle at 0.2 mm/s at 3dpf suffering a pressure drop of

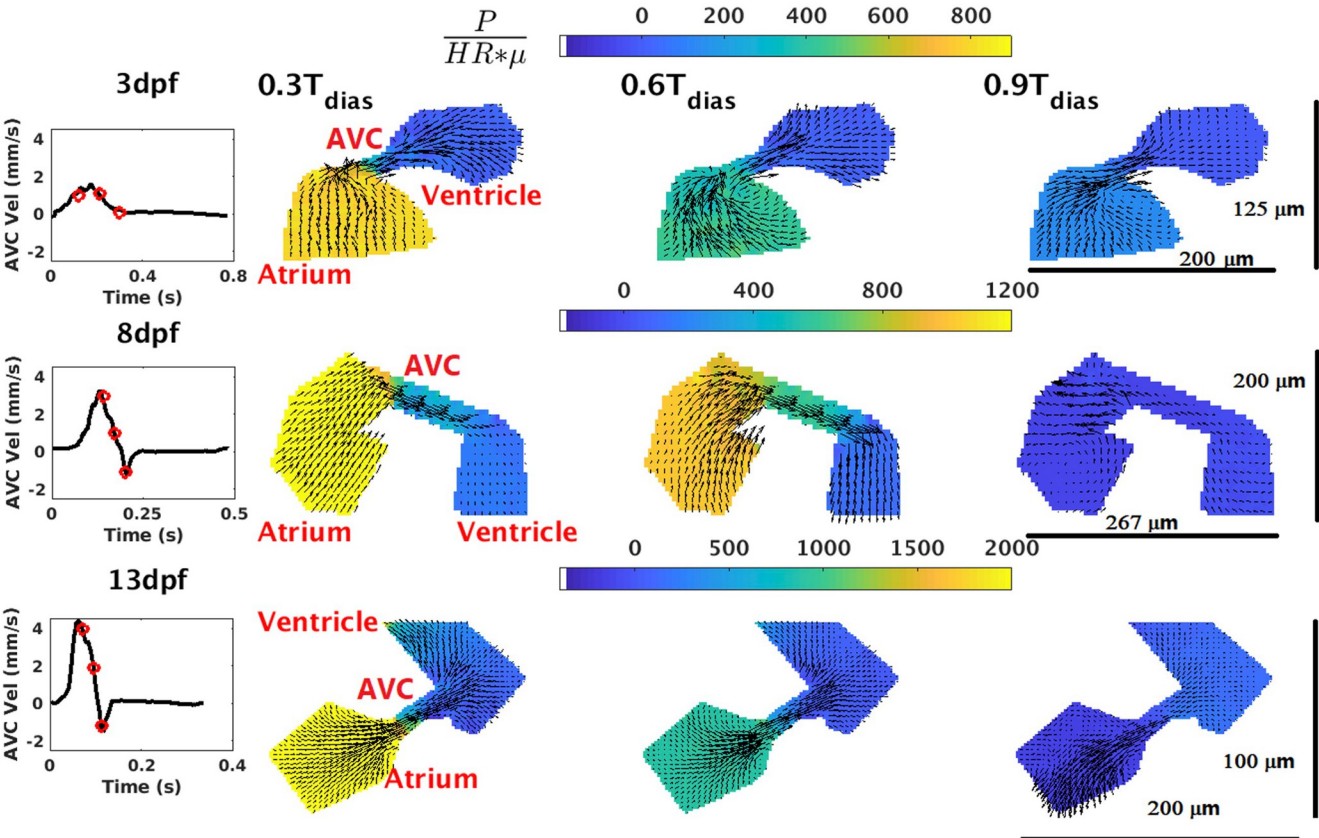

**Fig 3. Non-dimensional pressure field across ventricle, atrium, and AVC during ventricle diastole at 3 dpf, 8 dpf, and 13 dpf.** The black curve at each dpf in Column 1 shows the velocity profile at AVC (which is the inlet to the ventricle). The red dots on the black curve represent the $0.3T_{dias}$, $0.6T_{dias}$ and $0.9T_{dias}$ phases of the diastolic (ventricle-filling) cycle. The pressure contours and overlapping velocity vectors at $0.3T_{dias}$, $0.6T_{dias}$ and $0.9T_{dias}$ are respectively shown in Column 2, Column 3 and Column 4. Non-dimensional pressure at ventricle outflow of each pressure plot is assumed to be zero. Additionally, the spatial scales of the heart chambers at each dpf are shown in Column 4. Three rows of the figure represent data at 3 dpf, 8 dpf and 13 dpf respectively. The colors bars in each row represent the range of non-dimensional pressure in the heart chambers at each dpf.

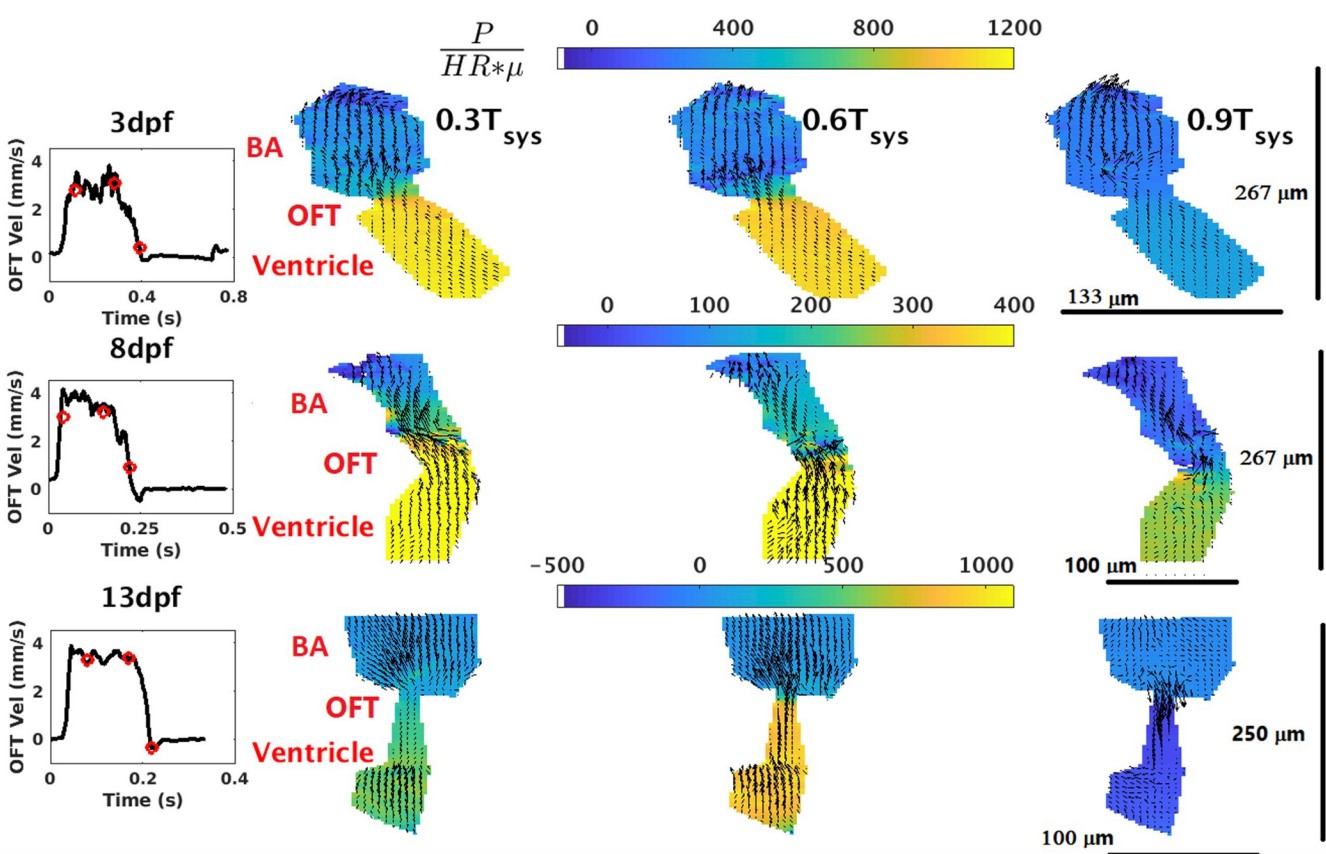

**Fig 4. Non-dimensional pressure field across ventricle, BA, and OFT during ventricle systole at 3 dpf, 8 dpf, and 13 dpf.** The black curve at each dpf in Column 1 shows the velocity profile at OFT (which is the outlet of the ventricle). The red dots on the black curve represent the $0.3T_{sys}$, $0.6T_{sys}$ and $0.9T_{sys}$ phases of the systolic (ventricle-outflow) cycle. The pressure contours and overlapping velocity vectors at $0.3T_{sys}$, $0.6T_{sys}$ and $0.9T_{sys}$ are respectively shown in Column 2, Column 3 and Column 4. Non-dimensional pressure at BA outflow of each pressure plot is assumed to be zero. Additionally, the spatial scales of the heart chambers at each dpf are shown in Column 4. Three rows of the figure represent data at 3 dpf, 8 dpf and 13 dpf respectively. The colors bars in each row represent the range of non-dimensional pressure in the heart chambers at each dpf.

200*(HR*μ) at the AVC region. However, due to the absence of the valves in the AVC, there is a negative pressure drop across the AVC that causes retrograde flow with a velocity of approximately 1 mm/s from the ventricle to atrium [-200*(HR*μ) at 8 dpf and -500*(HR*μ) at 13dpf] (Fig 3).

The pressure field variation across the ventricle and the bulbus arteriosus (BA) along three phases of the ventricle systolic cycle is plotted in Fig 4 for a medaka heart at 3 dpf, 8 dpf, and 13 dpf. A zero-pressure boundary condition is applied at the BA outflow. The narrow region joining the ventricle and the BA is referred to as the outflow tract (OFT). Both the atrial inflow and the AVC inflow to the ventricle remain closed throughout $T_{sys}$, which is the period of the ventricle systolic cycle. At $0.3T_{sys}$ the peak pressure drop across the OFT is 1200*(HR*μ) at 3dpf, 550*(HR*μ) at 8 dpf, and 700*(HR*μ) at 13dpf. The ventricle experiences higher pressure, and blood flows through the OFT from the ventricle to the BA. At $0.6T_{sys}$, the OFT pressure drop almost stays constant in magnitude during 3 dpf and 8dpf. [3dpf: 1150*(HR*μ), 8dpf: 500*(HR*μ)]. Interestingly for the 13 dpf heart, the OFT pressure drop increases substantially at $0.6T_{sys}$ to 1100*(HR*μ). At $0.9T_{sys}$ the ventricle outflow has almost ended with blood flowing from the ventricle to BA at 0.1 mm/s at 3dpf and 8dpf, suffering a pressure drop of 300*(HR*μ) at the OFT region. Nevertheless, due to the absence of the valves in the narrow

OFT, a negative pressure drop of -300*(HR*μ) at 13dpf causes retrograde flow with velocity 0.3 mm/s from the BA to the ventricle (Fig 4).

## Time variation of pressure drop across AVC and OFT

Time variations of velocity magnitude and pressure drop across AVC (dimensional and non-dimensional) are plotted for a ventricle diastolic cycle in Fig 5. Similar variations of velocity magnitude and pressure drop across OFT are plotted for a ventricle systolic cycle in Fig 6. The pink shaded region between the minimum and maximum time variations in each dpf depicts the bounds within which the profiles across 3–5 samples fluctuate. The black line shows the median profile for each dpf. The peak values of the median profile are marked on each plot by a blue circle.

A small pressure drop between 3dpf and 5dpf drives the low velocity flow into the ventricle when the AVC is not clearly defined. At 6dpf peak values of both $Vm_{AVC}$ and $\Delta P_{AVC}$ increase considerably owing to narrowing of the AVC region after cardiac looping. At 7dpf, the peak median $Vm_{AVC}$ and the peak median $\frac{\Delta P_{AVC}}{HR*\mu}$ reduce a little. As the fish prepares to hatch between 8 dpf and 9 dpf, there is a rise in peak flow velocity and pressure drop. After hatching, they eventually plateau from 10 dpf to 14 dpf. At 3 dpf, unlike the AVC, the peak flow velocity and pressure drop through OFT is high, which reduces at 4 dpf and 5 dpf. At 6dpf, there is a

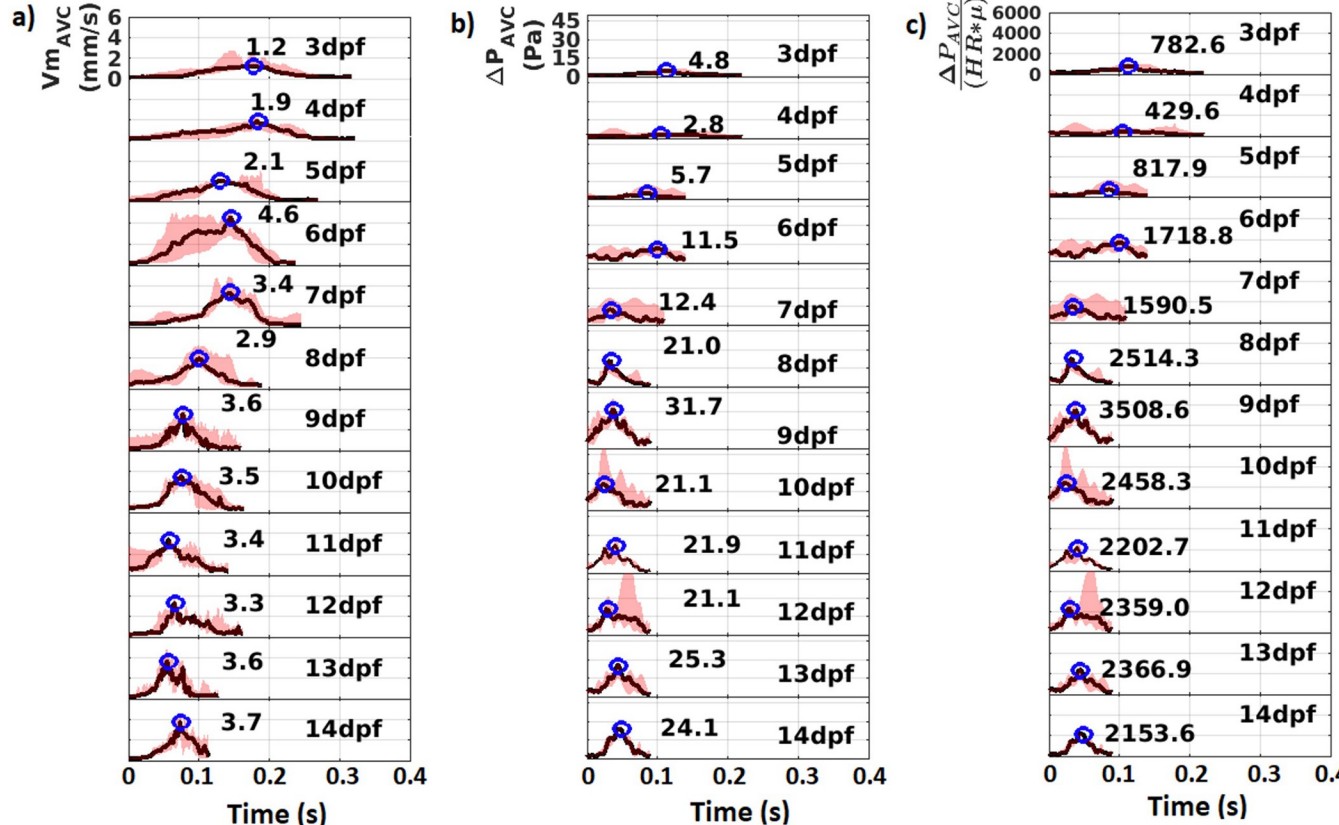

**Fig 5.** Time variation profiles in the atrioventricular canal (AVC) for the flow parameters a) Velocity Magnitude ($Vm_{AVC}$) b) Pressure drop ($\Delta P_{AVC}$) and c) Non-dimensionalized pressure drop $\frac{\Delta P_{AVC}}{HR*\mu}$. Black lines denote the median profiles while the pink shade denotes region between maximum and minimum profiles. The peak values of the median profile are marked on each plot by a blue circle. n = 3 for 8 and 11 dpf, n = 5 for all other dpf. For each n, 8 cycles are analyzed for 3 to 7 dpf samples and 10 cycles are analyzed for 8–14 dpf samples.

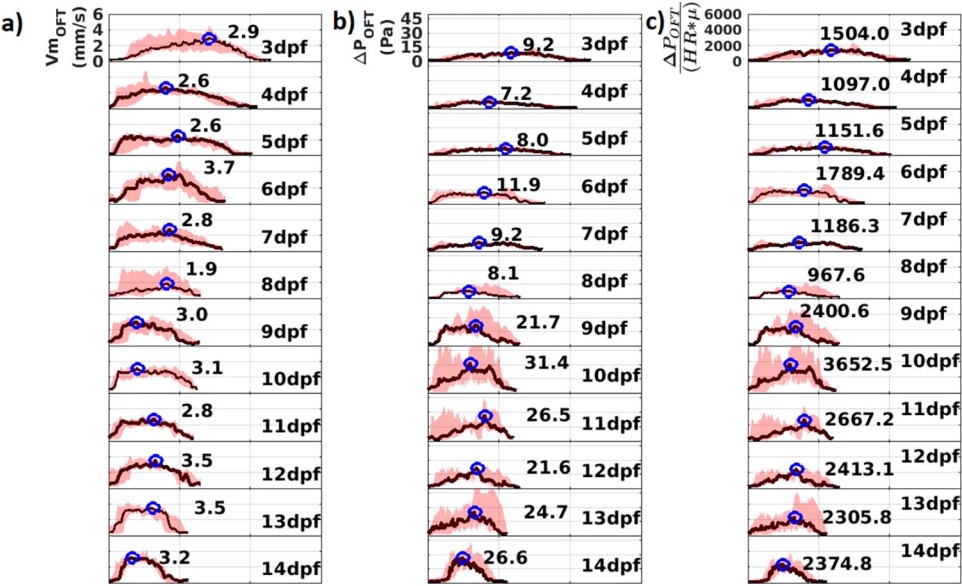

**Fig 6.** Time variation profiles in the atrioventricular canal (OFT) for the flow parameters a) Velocity Magnitude ($Vm_{OFT}$) b) Pressure drop ($\Delta P_{OFT}$) and c) Non-dimensionalized pressure drop $\frac{\Delta P_{OFT}}{HR*\mu}$. Black lines denote the median profiles while the pink shade denotes region between maximum and minimum profiles. The peak values of the median profile are marked on each plot by a blue circle. n = 3 for 8 and 11 dpf, n = 5 for all other dpf. For each n, 8 cycles are analyzed for 3 to 7 dpf samples and 10 cycles are analyzed for 8–14 dpf samples.

substantial rise in the peak median $Vm_{OFT}$. Moreover, the peak median $\frac{\Delta P_{OFT}}{HR*\mu}$ that can be attributed to the narrowing of OFT after cardiac looping. These parameters decrease subsequently at 7 dpf and 8 dpf. After hatching, the peak median $Vm_{AVC}$ and the peak median $\frac{\Delta P_{AVC}}{HR*\mu}$ increase at 9 dpf and fluctuates within a small range from 9 dpf to 14 dpf.

## Strain measurements of the ventricle wall

The strain is very low in the middle of the ventricle as compared to its value near the walls. So, in Fig 7, only strain values at the ventricle wall are shown. Wall strain as a metric has translational clinical potential. Hence, in this work we focus only on endocardial wall strain. Endocardial wall strain (%) is extracted during a heart cycle for each sample at each dpf. Fig 7A shows 3 dpf, 8 dpf and 13 dpf ventricle during early ($0.2T_{dias}$), peak ($0.5T_{dias}$) and late diastole ($0.8T_{dias}$). The wall locations on the endocardium are colored by the % strain experienced at each location. Fig 7B shows the time variation of endocardial strain along a heart cycle. A black line represents the median profile across 5 samples at each dpf. The pink shaded region between the minimum and maximum time variations in each dpf depicts the bounds within which the profiles across 5 samples fluctuate. The peak strain during ventricle diastole of the median profile is marked on each plot by a blue circle. The peak diastolic strain evolution with age progression is low at 3 dpf (19.5%) and 4 dpf (22%). But 5 dpf onwards, the median peak does not follow a specific trend with age progression. It fluctuates between 29% and 55%.

## EW measurement from $\Delta P_{AVC}$-strain loop

Pressure drop across AVC ($\Delta P_{AVC}$) is plotted across endocardial wall strain for a ventricle diastolic cycle at each dpf in Fig 8 for an individual subject medaka heart. The endocardial work (EW) represents the area under this $\Delta P_{AVC}$-strain curve. EW obtained from this will have the

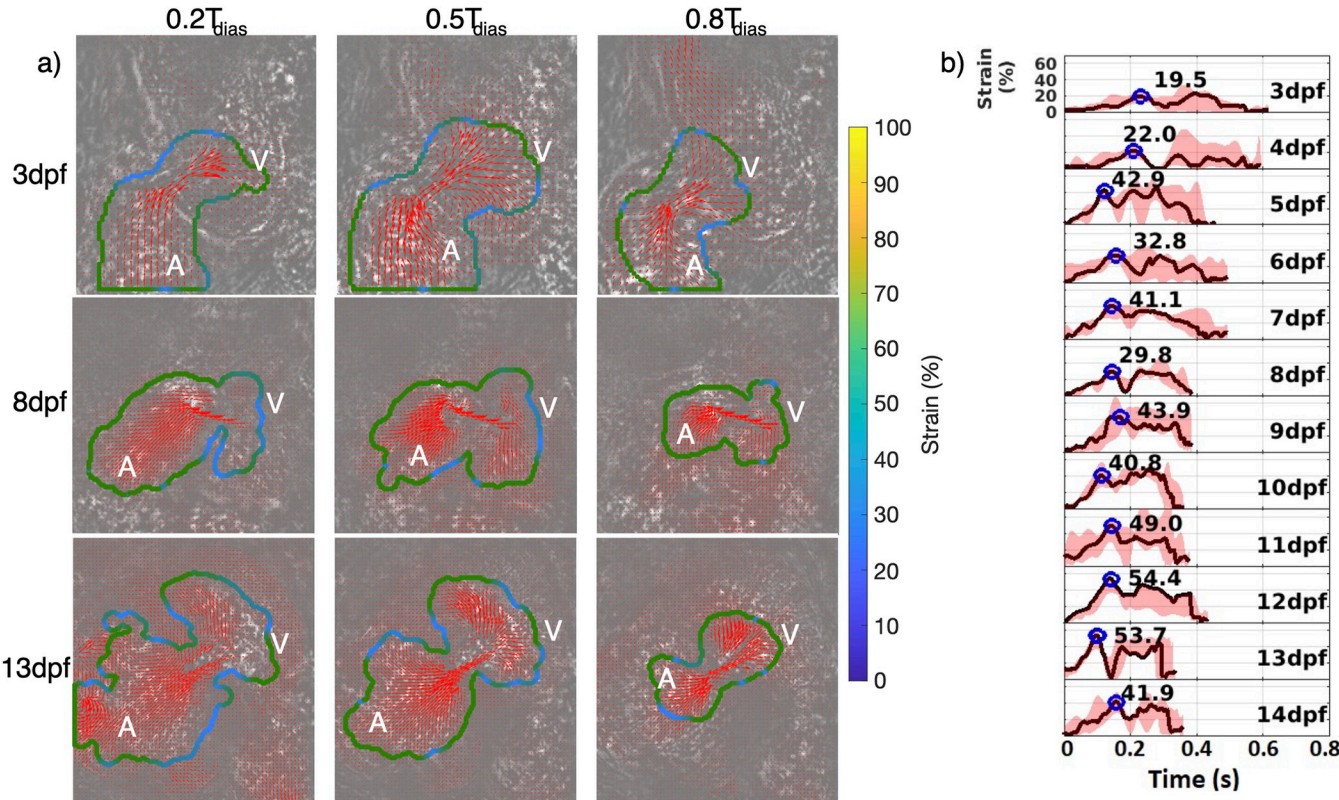

**Fig 7. Strain (%) measurement on ventricle wall and its temporal evolution along a medaka heart cycle at different age progression.** a) Endocardial strain overlapped with velocity vectors on the raw images for 3, 8 and 13 dpf medaka hearts at early ($0.2T_{dias}$), peak($0.5T_{dias}$) and late diastolic($0.8T_{dias}$) filling of the ventricle. A stands for atrium and V stands for ventricle. b) Variation of endocardial strain (%)-time history for each heart cycle across medaka age progression. Blue circles on each curve at each dpf represent the maximum endocardial strain (%) during peak diastolic filling of the ventricle.

unit Pa-% instead of the conventional unit of N-m. This EW parameter is associated with the myocardial work (MW) analysis, which looks at the relationship between cuff-based blood pressure and ventricular global strain. The MW parameter is a novel quantity, becoming popularized in cardiology and echocardiography, for its ability to capture changes in ventricular filling function as a function of relaxation (deformation and strain) due to cardiac remodeling [4,44]. We are therefore looking at EW in this work to understand how changes in ventricular remodeling due to growth affect filling pressures.

The three curves in each subplot are selected according to a minimum (earliest dpf), maximum (latest dpf), and median EW (middle dpf). The axis limits for each subplot are identical to allow for qualitative comparison between ages. From 3 dpf to 5 dpf, a low area under the curve exists as filling pressures are not elevated. Across cases, strain is not significantly altered, but shows reduced endocardial wall contractility compared to the larger area curves (6dpf to 14 dpf). Furthermore, the figure suggests that for the cases observed, pressure difference does not significantly change until the medaka fish reach hatching age, at which point breathing and function must be sustained solely by the creature. The maximum strain occurs when the ventricle expansion is the largest, while the maximum pressure drop occurs after the peak flow velocity crosses the AVC.

The curve shapes are noisy and inconsistent owing to two reasons: a) the reference (end of ventricle systole) time may have been slightly different in each sample. b) the velocity gradients calculated from the vectors of μPIV analysis are susceptible to 10% error for 1% error in

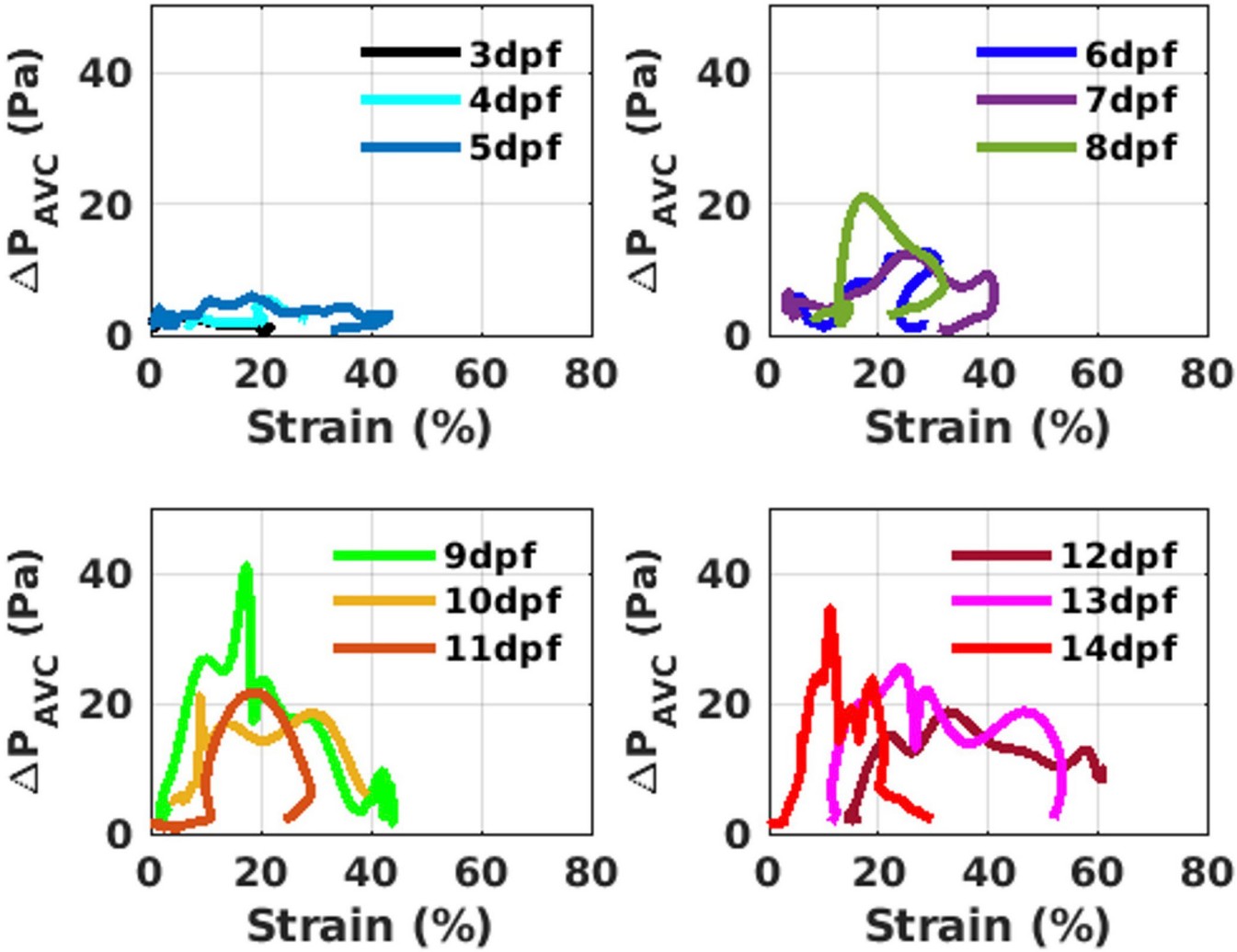

**Fig 8. Pressure drop across AVC ($\Delta P_{AVC}$) vs. endocardial wall peak strain at each time point during ventricle diastole from 3 dpf to 14 dpf.** Instantaneous $\Delta P_{AVC}$ vs strain plot extracted from one sample at each dpf. The y-axis and x-axis limits for all subplots are the same. The area under the curve visibly increases considerably from the pre-hatch period (3 dpf-8 dpf) to post hatch period (9 dpf -14 dpf) of the medaka.

velocity measurements [35,40]. These errors may show up as "noisy" plots. Furthermore, only one individual is presented per curve, which was done because direct comparison across all subjects at each dpf was not suitable. This is due to differences in diastole onset time and duration between subjects. This prevents statistical analysis between curves for all individuals.

## Peak EW, peak ΔP at AVC, OFT, and peak diastolic strain measurement

The peak values of EW, $\Delta P_{AVC}$, $\Delta P_{OFT}$, and diastolic strain have been plotted against developing medaka age progression in Fig 9. Statistical analysis of these values between dpf are provided in S1 Table. The mean across 4–5 samples in each dpf is plotted, with the standard deviation being displayed by the error bars. The (\*\*\*) over each dpf shows that the mean is statistically significantly different when compared to the mean value at 3 dpf. In Fig 9D, the peak strain during ventricle diastole ($Strain_{dias}$) is low at 3dpf and 4 dpf. It fluctuates within a broad range from 5 dpf and 14 dpf, with high variability in each dpf. So, the strain metric cannot accurately distinguish ventricle wall remodeling stages. In Fig 9B and 9C, peak values of

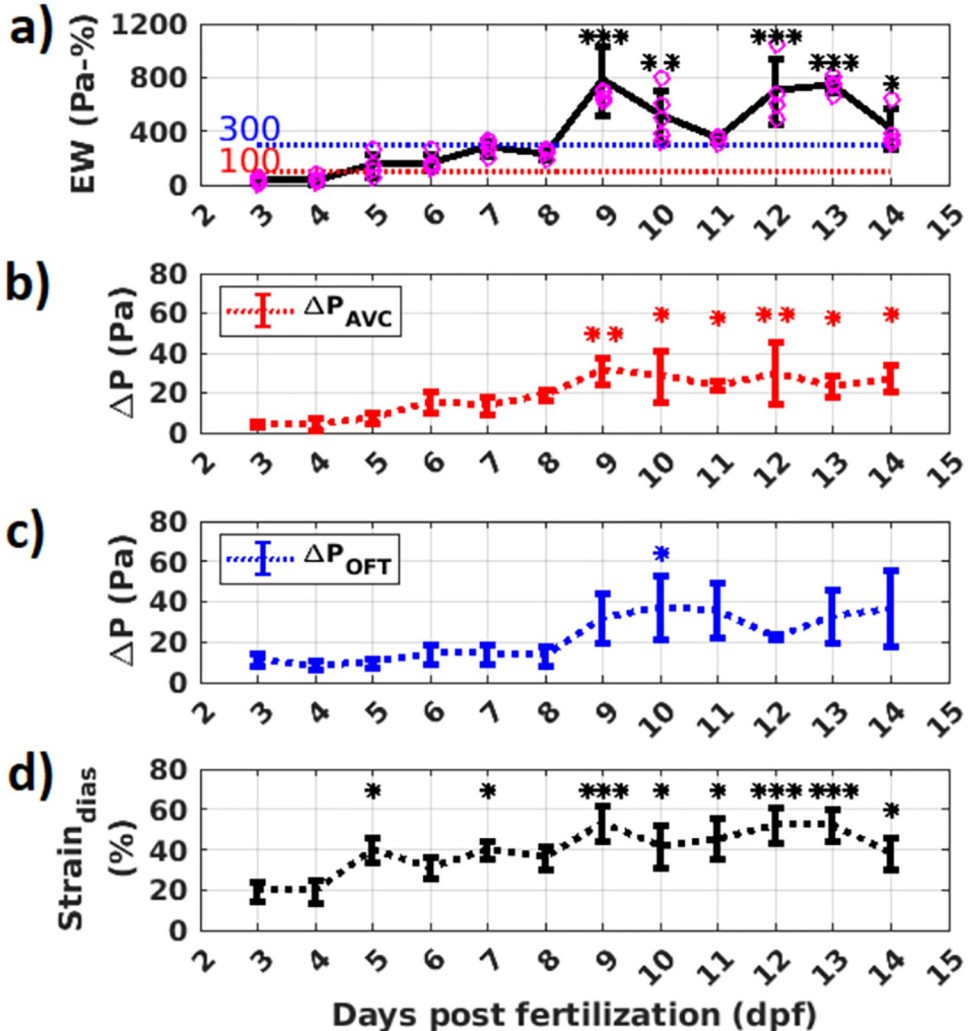

**Fig 9.** Variation of a) Peak EW, b) peak $\Delta P$ at AVC c) peak $\Delta P$ at OFT, and d) peak diastolic ventricle strain with medaka age progression. Age is represented in terms of days post fertilization (dpf). The * in the plots denote statistical significance compared to the control values in 3 dpf. *** denote $p<0.0001$, ** denote $p<0.001$, and * denote $p < 0.05$.

$\Delta P_{AVC}$ and $\Delta P_{OFT}$ show a well-defined increase from pre-hatch (3dpf—8 dpf) to post-hatch stages (9 dpf– 14 dpf). The variability across the samples is high in the post-hatch stages. Fig 9A shows a well-defined progression of EW with developing fish age. EW at 3 dpf and 4 dpf are below 100 Pa-%. EW at 5 dpf and 6 dpf are below 300 Pa-%, while the EW at 9 dpf, 10 dpf, 12 dpf, and 13 dpf are above 300 Pa-%. The sudden reduction of EW and $\Delta P_{AVC}$ at 11 dpf with little variability can be attributed to unknown developmental changes occurring at AVC. A larger sample size is required to substantiate this observation. Overall, EW is the metric that best demarcates between the linear heart stage ($< 100$ Pa-%), cardiac looped chamber stage ($< 300$ Pa-%), and the fully formed chamber stage ($> 300$ Pa-%).

## Discussion

Non-invasive pressure drop measurements are reported for the first time across the AVC and OFT regions of a developing medaka during the linear heart tube stages, cardiac looping end stages, and fully developed chamber stages. Retrograde flows were observed in pressure

contour and velocity vector plots at AVC during 8 dpf and 13 dpf and at OFT during 13 dpf (Figs 3 and 4). The pumping mechanics at the AVC and OFT during 3dpf are significantly different, as observed from the pressure contours. The AVC is not narrow at 3dpf, due to which the pressure drop across the chambers is small. With age progression, the AVC starts narrowing, and the $\Delta P_{AVC}$ increases. The OFT is initially positioned such that during ventricle systole, a strong suction force draws blood directly from the atrial inlet. Due to this large $\Delta P_{OFT}$, a high flow velocity is observed at 3dpf. At 6dpf, the appreciable increase in both $\Delta P_{AVC}$ and $\Delta P_{OFT}$ can be attributed to similar filling mechanics of flow into a compressed closed vessel, like a balloon. This increase was observed in peak-WSS values in the previous work with this data [18]. The balloon-filling mechanics continue in the successive embryonic ages. After hatching, $\Delta P_{AVC}$ and $\Delta P_{OFT}$ are appreciably higher than their pre-hatch measures. Due to the decompression of the organs, the ventricle rapidly increases in size after hatching. This leads to a stronger pressure drop when the ventricle expands ($\Delta P_{AVC}$) or contracts ($\Delta P_{OFT}$). The non-dimensional pressure drop is useful to compare across stages within the same species and with other species where the length and time scales are different [14]. The pressure measurement technique can also be used for three-dimensional images. When combined with physical pressure readings, this can predict the absolute pressure distribution in the medaka heart in future studies.

Strain measurements have not been explored earlier in medaka heart. Our measurement technique avoids the lack of repeatability associated with the manual segmentation of cardiac walls. Instead, using velocity gradients to detect wall edges circumvents the expertise required to identify endocardium in a strain calculated from ultrasound scans. The results show that the ventricle deformation is much less in the linear heart tube stage at 3 dpf and 4 dpf. This is attributed to the lack of cardiomyocytes that start populating significantly after 5 dpf [45]. The peak diastolic strain measured earlier in the normal zebrafish heart was around 40% at 50 hpf, 75 hpf, and 100 hpf [3] which could not differentiate between the three developmental stages. The %-strain is still a useful metric to compare the contractility in different segments of the cardiac wall and can identify disease regions on the cardiac wall.

The EW is a metric that we have introduced through this work in the teleost cardiac mechanics literature. The EW has shown sufficient demarcation (Fig 9A) between the three main developmental stages of the medaka heart but needs to be validated with a higher number of samples at each dpf. A lower EW suggests a stiffer ventricle with low flowrates through the AVC. The shape of the $\Delta P_{AVC}$-strain curve at different stages offers insight into the pumping mechanism and time scales of ventricle expansion, relaxation, and contraction. These can be quantified in future studies with more samples. A flat-topped curve (Fig 6) indicates that the $\Delta P_{AVC}$ is caused by the positive pressure in the atrium only with no contribution from the ventricle expansion. A bell-shaped curve, on the other hand, includes the contribution of ventricle expansion to the $\Delta P_{AVC}$. The EW calculated from these curves is indirectly related to the metabolic demand of the heart and can be further developed into a diagnostic tool that can differentiate normal hearts from diseased hearts in humans.

## Limitations

The main limitation of this study is that there are only five samples per dpf. Including more samples would require the investment of sufficient time and resources but would increase accuracy. However, multi-cycle (8–10) recording per sample with high spatiotemporal resolution increases the effective number of correlating pixels (RBS pattern area times number of correlating RBC patterns times number of cycle points contributing in phase averaging) significantly. This combined with advanced ensemble correlation processing with phase averaging

reduces the velocity measurement uncertainty leading to more accurate and statistically converged measurements. We did a statistical power analysis considering a 2-sided group design at a 0.05 alpha level and with sample size 3 for each dpf. We compared means of a study group with the expected mean of a known population at each dpf and the resulting power came out to be between 75% and 80%. This value is acceptable to detect a difference between the two groups.

The limited spatial resolution near the endocardial wall introduced errors in the velocity gradient measurements that propagated in both pressure and strain calculations. The orientation of the heart during imaging can be different for different samples at each dpf. This limits a point-by-point comparison of the instantaneous wall locations and flow field measurements across samples tested at each dpf, however, the peak strain, pressure and EW measurements would not be affected by the change in orientation. In some orientations, the out-of-plane-velocity vectors are larger in number at the AVC. The decrease in correlation peak height due to out-of-plane motion is minimized by ensemble phase correlation process. Multi-pass iterative deform processing with validation also reduce invalid measurements due to signal dropout. However, any remaining error may bias the final pressure and strain calculations. The EW is calculated from an open $\Delta P_{AVC}$-strain loop because the pressure cannot be calculated when the AVC is closed. Despite all these limitations, both the pressure drop at AVC, OFT, and the EW show a gradual increase with developing medaka age progression that has not been reported yet. Thus, state-of-the-art processing, a higher number of effective observations, and improved accuracy minimize intra-sample variability. Though this does not directly account for inter-sample variation, reduction in intra-sample uncertainty aids in observing statistical significance for EW (before and after 8 dpf) across the 3–5 samples analyzed in this work.

## Conclusion

To our knowledge, this study is the first to report peak pressure drop values in the AVC and OFT of a developing Japanese medaka from 3 dpf (onset of blood circulation) to 14 dpf. The viscosity used in the pressure calculations was calculated at each dpf instead of assuming a constant value. Flow reversals observed for the first time in the valveless narrow cushion regions of the medaka ventricle inflow and outflow are validated by negative pressure drop towards the end of ventricle diastole during 8 dpf and 13 dpf. A non-intrusive metric EW is introduced to differentiate between the linear heart tube stages, end of cardiac looping stages, and the fully developed chambers in the post-hatch stages.

Longitudinal studies of cardiovascular development relating the pressure drop to cardiac developmental landmarks are scarce in all vertebrates. An important insight gained from this longitudinal analysis of flow and tissue mechanics variation with age is the different pumping mechanisms of the heart at different dpf. At 3dpf, the pumping mechanics at the AVC was different from that of the OFT. This baseline framework can be further tested by probing the flow-tissue mechanics evolution in higher vertebrate models.

Overall, a consistent increase in pressure drop, and higher EW during embryonic heart development is a measure of narrowing AVC and increasing wall contractility associated with cardiac remodeling. The EW can also be correlated with the metabolic demand of the heart. A deviation in the measured metric could indicate abnormal heart development. The automated advanced boundary segmentation and flow processing algorithm increases the reliability of the measured metrics. The non-invasive hydrodynamic measurements estimated using our novel workflow can be further developed as a diagnostic tool to differentiate between normal and diseased hearts.

## Supporting information

**S1 Fig. A schematic of the PIV image processing steps.** Each instantaneous image sequence is divided into windows and the same windows for two consecutive images are used for processing. A 50% Gaussian filtering is applied on each window and the filtered windows are cross-correlated using SCC to get instantaneous shift estimates. This process is followed for image pairs in the same phase for different cycles. The SCC planes from different cycles are averaged to obtain an ensemble SCC plane. A subpixel fit is done to estimate the peak location which denotes the shift between image pairs. This process is repeated for all windows in the image. An outlier detection is performed on the vector field. The whole process is repeated for multi-pass processing with the estimated field used to deform the original image windows. (PDF)

**S1 Table. p-values from Tukey Kramer HSD tests.** The table compares Peak Strain$_{dias}$, peak $\Delta P_{AVC}$, peak $\Delta P_{OFT}$, peak $\frac{\Delta P_{AVC}}{HR*\mu}$, Peak $\frac{\Delta P_{OFT}}{HR*\mu}$ and EW between each pair of dpfs. n = 5 at each dpf. *** denotes p<0.0001. ** denotes p<0.001. * denotes p<0.05. (PDF)

## Author Contributions

**Conceptualization:** Sreyashi Chakraborty, Maria S. Sepúlveda, Pavlos P. Vlachos.

**Data curation:** Sreyashi Chakraborty.

**Formal analysis:** Sreyashi Chakraborty, Sayantan Bhattacharya.

**Supervision:** Maria S. Sepúlveda, Pavlos P. Vlachos.

**Writing – original draft:** Sreyashi Chakraborty.

**Writing – review & editing:** Sreyashi Chakraborty, Sayantan Bhattacharya, Brett Albert Meyers, Maria S. Sepúlveda, Pavlos P. Vlachos.

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
