## [Decision Letter · Decision Letter 0]

8 Apr 2024

PONE-D-23-31127Evolution of cardiac tissue and flow mechanics in developing Japanese MedakaPLOS ONE

Dear Dr. Vlachos,

Thank you for submitting your manuscript to PLOS ONE. After careful consideration, we feel that it has merit but does not fully meet PLOS ONE’s publication criteria as it currently stands. Therefore, we invite you to submit a revised version of the manuscript that addresses the points raised during the review process.

My sincere apologies for the slow progress of this review. It took a lot longer than expected to recruit appropriately qualified reviewers who had the time to contribute to the review process.As you will see, both reviewers have raised some critical points that should be addressed completely before we reconsider this manuscript. Essentially there are some errors and missing details and also there is a general need to clarify the presentation of central experimental approaches. Also, I would recommend that the authors pay some attention to resolving one key comment of Reviewer 1, namely the need to explain in more detail, how this manuscript goes beyond just being a descriptive piece of work and instead generally improves our understanding of embryonic heart development. Also, how may the methodology that has been used help future studies in this field.In other words, the authors need to spend a little bit more effort to “sell” the importance of this work for the field…

We look forward to receiving your revised manuscript.

Kind regards,

Nicholas Simon Foulkes, D.Phil

Academic Editor

PLOS ONE

Journal Requirements:

Reviewers' comments:

Reviewer's Responses to Questions

**Comments to the Author**

1. Is the manuscript technically sound, and do the data support the conclusions?

Reviewer #1: Yes

Reviewer #2: Partly

2. Has the statistical analysis been performed appropriately and rigorously? 

Reviewer #1: I Don't Know

Reviewer #2: Yes

3. Have the authors made all data underlying the findings in their manuscript fully available?

Reviewer #1: Yes

Reviewer #2: Yes

4. Is the manuscript presented in an intelligible fashion and written in standard English?

Reviewer #1: Yes

Reviewer #2: Yes

5. Review Comments to the Author

Reviewer #1: Evolution of cardiac tissue and flow mechanics in developing Japanese Medaka

Authors: Sreyashi Chakraborty1, Sayantan Bhattacharya2, Brett A. Meyers1, Maria S. Sepulveda3, Pavlos Vlachos1,4*

The manuscript "Evolution of cardiac tissue and flow mechanics in developing Japanese Medaka” uses image analysis of image sequences obtained with high speed cameras to calculate blood flow and pressure in the developing medaka heart. The presented data provide interesting parameter analysis during embryonic heart development. The manuscript is of mainly descriptive nature reporting a specialized methodology without reaching novel conclusions on heart development or function. The authors should provide a conclusion why their results are relevant for a study of embryonic heart development, and explain how their work goes beyond being mostly descriptive. It would be good to provide an outlook on how they envision that their methodology will contribute to an analysis and understanding of heart development and function. Nevertheless this work is of interest for the wide spectrum of readers of PlosOne and thus merits publication after some revision.

minor points

line 91: The authors should specify the strain used for this study. Several highly inbred strains are available for medaka. Use of an inbred strain allows to keep variance during analysis low which is important for acquisition of quantitative data.

line 92: please give details what “continuous oxygen supply”means. Does this refer to continuous water circulation?

line 100: were any measures taken to improve transparency of the chorion or rather reduce chorion induced image distortions when imaging embryos prior to hatching?

lines 103 ff: the imaging methodology of unhatched and hatched embryos/hatchling has to be be described better: in which medium were the embryos immersed during imaging? Was a climate chamber used to keep T constant? Were hatchlings anesthetised during the imaging to prevent moving. How was a imaging of a lateral view achieved in the actively swimming hatchlings?

line 150 the terms AVC and OFT need an introduction here for the understanding

line 195: give a definition of hematocrit for the understanding

line 278: here is the place to use the term diastole together with “ventricle filling cycle”.

Reviewer #2: The provided manuscript presents a study of the evolution of a set of biomechanical characteristics during cardiac morphogenesis in the Japanese Medaka. Using microscopy particle image velocimetry quantifications, 2D viscous fluid velocity fields in the valve regions were reconstructed; based on which pressure fields and endocardial wall strains at several developmental time points were calculated. The authors demonstrated that pressure drops, and the relationship between pressure and strain can be used to distinguish developmental time points.

L27-L28: The authors mention that “The effects of pressure drop [...] triggering cardiac morphogenesis in a teleost heart species are poorly understood”, but they do not further consider this in their manuscript. As such, it is misleading and should be altered.

L40-L42: Missing reference(s)

L42-L45: Missing reference(s)

L47-L49: Missing reference(s)

L58-L59: “viscous

L61-L62: “[...] because the [opacity of the] fish’s eyes and head blocked [visibility of] a part of the ventricle.”

L95: Missing concentration of diluted saline solution

L115: “RBC” - abbreviation not previously defined/introduced

L119-L122; Figure 1: Scale bars are missing in all panels

L124: “HR” - abbreviation not previously defined/introduced

L126: Missing reference for parameter value (blood density)

L127: “EF” - abbreviation not previously defined/introduced

Equation 1: The assumption of cylindrical ventricle area and volume appears to be unjustified, as its shape deviates from tubular already one day post-fertilization (1dpf). The authors themselves show in their Figure 2b-d sketches of non-cylindrical ventricles.

L131-L156: A visualization of the image processing steps would have been beneficial for the readers to understand the effect of the filtering.

L157-L165; Figure 2: On which sources of information are the sketches based?

L203: “ [...] as a function of hematocrit [percentage] (Ht[%]) [...]”

Equation 7: Not referenced in the text; Δ is not introduced; the first two terms on the right-hand side of equation 6 mentioned in L220-L221 as being negligible are still present.

L226: “Writing in terms of the [dimensions] [...]”

L229; Equation 10: Should add P= M/(LTT), = M/(LT) and insert into equation 10 to show dimensions vanish.

6. PLOS authors have the option to publish the peer review history of their article (what does this mean?). If published, this will include your full peer review and any attached files.

Reviewer #1: No

Reviewer #2: No

---

## [Author Response · Author response to Decision Letter 0]

28 Jul 2024

Response to the reviewer comments is attached. How the current work aids in the embryonic heart development and the applicability of the methodology is clarified in the conclusion.

---

## [Editor Report · Decision Letter 1]

5 Aug 2024

Evolution of cardiac tissue and flow mechanics in developing Japanese Medaka

PONE-D-23-31127R1

Dear Dr. Vlachos,

We’re pleased to inform you that your manuscript has been judged scientifically suitable for publication and will be formally accepted for publication once it meets all outstanding technical requirements.

Kind regards,

Nicholas S. Foulkes, D.Phil

Academic Editor

PLOS ONE
---

## [Editor Report · Acceptance letter]

15 Aug 2024

PONE-D-23-31127R1 

PLOS ONE

Dear Dr. Vlachos, 

I'm pleased to inform you that your manuscript has been deemed suitable for publication in PLOS ONE. Congratulations! Your manuscript is now being handed over to our production team.

Kind regards, 

on behalf of

Dr. Nicholas S. Foulkes 

Academic Editor

PLOS ONE